# Effects of Vegetable Fields on the Spatial Distribution Patterns of Metal(loid)s in Soils Based on GIS and Moran’s I

**DOI:** 10.3390/ijerph16214095

**Published:** 2019-10-24

**Authors:** Qiang Wang, Shanlian Yang, Menglei Zheng, Fengxiang Han, Youhua Ma

**Affiliations:** 1School of Resources and Environment, Anhui Agricultural University, Hefei 230036, China; 28104@ahau.edu.cn (Q.W.); 13721104864@163.com (S.Y.); zyzdzml@163.com (M.Z.); 2Department of Chemistry and Biochemistry, Jackson State University, Jackson, MS 39056, USA; fengxiang.han@jsums.edu; 3Institute for New Rural Development, Anhui Agricultural University, Hefei 230036, China

**Keywords:** spatial correlation, geoda, spatial variation, geostatistical, metal(loid)s

## Abstract

Metal(loid) pollution in vegetable field soils has become increasingly severe and affects the safety of vegetable crops. Research in China has mainly focused on greenhouse vegetables (GV), while open field vegetables (OV) and the spatial distribution patterns of metal(loid)s in the surrounding soils have rarely been assessed. In the present study, spatial analysis methods combining Geographic Information Systems (GIS) and Moran’s I were applied to analyze the effects of vegetable fields on metal(loid) accumulation in soils. Overall, vegetable fields affected the spatial distribution of metal(loid)s in soils. In long-term vegetable production, the use of large amounts of organic fertilizer led to the bioconcentration of cadmium (Cd) and mercury (Hg), and long-term fertilization resulted in a significant pH decrease and consequent transformation and migration of chromium (Cr), lead (Pb), and arsenic (As). Thus, OV fields with a long history of planting had lower average pH and Cd, and higher average As, Cr, Hg, and Pb than GV fields, reached 0.93%, 10.1%, 5.8%, 3.0%, 80.8%, and 0.43% respectively. Due to the migration and transformation of metal(loid)s in OV soils, these should be further investigated regarding their abilities to reduce the accumulation of metal(loid)s in soils and protect the quality of the cultivated land.

## 1. Introduction

The term “metal(loid)s” represents both metals and metalloids. Metal(loid) accumulation can lead to the contamination of surface water, groundwater, organisms, sediments, and oceans. Metal(loid) pollution in agricultural soils has become an urgent issue worldwide. This is of particular concern in China due to its rapid economic growth over the past 40 years, and metal(loid) pollution in soils is now being considered of high-risk to the environment and human health [1]; the consequential environmental problems have also received widespread attention [2]. Some researchers have employed multivariate statistical analysis and geostatistical analysis for qualitative or quantitative research to determine the heavy metal sources in soils; however, it is generally believed that natural sources and human activities are the two major providers of heavy metals [3,4,5]. Natural sources include rock components [6,7,8], soil parent material [9], and atmospheric sediments from soil formation processes, and these are concentrated in soils after weathering and leaching, attaining high geological background values. Human activities mainly comprise industrial activities such as mining, industrial emissions, coal combustion, point source emissions [10,11,12,13,14,15], agricultural production from the long-term and massive application of fertilizers [16,17], and life activities.

Vegetables from agricultural production are essential. However, the excessive use of chemicals to promote vegetable growth may lead to an increase in the concentrations of metal(loid)s in soils, which threatens human health through the food chain and should be investigated further. Due to market demand and economic incentives, large-scale agricultural greenhouse vegetable (GV) production is rapidly expanding worldwide, especially in developing countries [18,19,20]. Under such circumstances, research has begun to link GV production with trace metal accumulation in soils [7,21]. China is the world’s largest producer and consumer of vegetables, with the planting area and production both accounting for more than 40% of the global totals [22]. In China, intensive planting of vegetables involves the massive use of fertilizer for the purpose of high yields. The traditional vegetable planting management mode, i.e., “more fertilizer, higher yields,” leads to soil nutrient bioconcentration and metal(loid) accumulation, thereby causing soil acidification, salinization, and groundwater pollution. This results in the biological activity and decomposing ability of soils being reduced, leading to the deterioration of vegetable quality and to severe economic and environmental impacts [23]. According to the survey data of the National Bureau of Statistics of China and to published literature, the total area for vegetable cultivation in China was 19.981 million hectares in 2017 [24]; the GV production area reached 3.7 million hectares [25], while the open vegetable fields still accounted for 81.5% of the total vegetable cultivation area in China, reaching 16.3 million hectares. Although some studies have reported the spatial distribution [26], time variation [27], and environmental quality assessment [19] of trace metals in greenhouse vegetables in China, vegetables from open fields are more likely to be contaminated than those grown in greenhouses, as without any greenhouse or film protection open fields are more exposed to the migration and transformation of soil attributes such as nutrients and metal(loid)s, which greatly affect the soil environment. However, little research has been done to analyze the effects of open vegetable (OV) fields on the spatial distribution patterns of metal(loid)s in the surrounding soils. Therefore, whether OV fields are the pollution sources or sinks of metal(loid)s in soils of the surrounding land is unknown.

In the present study, Feidong County, Hefei, Anhui was selected as the subject area because it has witnessed rapid economic development, has a large agricultural area, of which a large portion is for vegetable production, and has no record of pollution sources. The object-oriented method was used to perform remote-sensing classification of the GV fields in this area and to obtain their spatial distribution in 2019. The official agricultural survey data of Feidong County were used to obtain the spatial distribution of vegetable fields in this area in 2018. There were 375 surface soil samples obtained from the agricultural pollution survey of the County. The spatial interpolation method was used to simulate the spatial distributions of one metalloid, arsenic (As), and the four heavy metals including cadmium (Cd), chromium (Cr), mercury (Hg), and lead (Pb). Geographic Information Systems (GIS) and local Moran’s I were used to analyze whether vegetable fields had an effect on the spatial distribution patterns of metal(loid)s in soils, whether the concentrations of metal(loid)s in agricultural soils increased due to the excessive use of fertilizers, and whether OV fields and protected agriculture led to the bioconcentration of metal(loid)s in the surrounding soils. This study provides data to support the management and protection of agricultural soil quality and toxicity, and it is of great scientific and practical significance.

## 2. Materials and Methods

### 2.1. Study Area

Feidong County, located in the central part of Anhui Province (Figure 1) and on the east side of Hefei (the capital of Anhui Province), is one of the most important areas for vegetable agriculture in Hefei. The total area of the county is about 221,200 hectares, most of which belongs to the Chaohu Lake Basin. The cultivated land area is about 76,700 hectares, accounting for 35% of the total area. The total population is about 1.1 million, of which 970,000 are employed by the agricultural industry, evidencing that Feidong is an agricultural county. According to the morphological characteristics, the county can be divided into four areas: low mountainous area in the east, low hilly area in the north, wavy plain area in the middle, and lakeside plain area in the south. Feidong County belongs to the north subtropical monsoon climate zone, with abundant sunshine, mild climate, four distinct seasons, and moderate rainfall. Paddy soils and yellow cinnamon soils are the two primary soil types, accounting for 72.72% and 20.33% of the area, respectively (Table 1 and Figure 2). The total area of vegetable fields (including OV and GV fields) was 6159.126 hectares. The soil type data and vegetable field data were obtained from the Anhui Provincial Agriculture Committee.

From July to December 2017, 375 0–20 cm topsoil samples were collected across Feidong County (Figure 1) using bamboo shovels. During the sampling, areas with new and locally contaminated soils were avoided, and the geographic coordinates of the sampling sites were recorded. After sampling, the GPS-measured sampling points with coordinate records were converted into points with spatial coordinates using ArcGIS 10.2 (Environmental Systems Research Institute, Inc, RedLands, CA, USA) and projection transformation was applied to generate a sample distribution map with soil metal(loid) information.

### 2.2. Data Sources and Technical Paths

#### 2.2.1. Vegetable Field Types

Among the vegetable field types, the greenhouse vegetables are an area covered with a greenhouse facility. The location and area of greenhouse facilities in 2019 were extracted by the four spatial attributes of area, length, rectangular_fit, and major_Length with the Rule-based Classification step in the ENVI software from the Google Earth Level 17 data, as shown on the left and middle in Figure 3. The overall accuracy was 77.439% and the kappa coefficient was 0.618, then the extraction results were modified manually. The open field vegetables are the vegetable fields without greenhouse facilities minus the greenhouse vegetable areas, to reduce the impact of different types of vegetable fields, as shown on the right in Figure 3.

The obtained spatial distribution was intersected with the spatial distribution of the total vegetable fields from Feidong’s agricultural survey data in 2016 to obtain the spatial distributions of the OV and GV fields (Figure 4).

#### 2.2.2. Soil Sampling and Analysis

The soil tests were completed at the Anhui Institute of Geological Experiment. After the samples were digested by HCl–HNO_3_–HClO_4_–HF, the concentrations of Cd, Pb, and Cr were determined by inductively coupled plasma mass spectrometry (ICP-MS 7700, Agilent Technologies, Santa Clara, CA, USA); the concentrations of Hg and As were determined by atomic fluorescence spectrometry (AFS-8220, JiTian Technologies, Beijing, China) after digestion by aqua regia; the soil pH was measured using a pH meter (SG8, METTLER TOLEDO, Zurich, Switzerland). The laboratory used national standard materials and repeated sample tests for quality monitoring. The accuracy, precision, and reported percent of the various indicators were controlled at 0.10–0.12, 10–20%, and over 98%, respectively.

### 2.3. Geostatistical Interpolation and Kernel Density Simulation

The ARCMAP10.2 platform (Environmental Systems Research Institute, Redlands, CA, USA), designed by the Environmental Systems Research Institute (ESRI), was used for the geostatistical interpolation of metal(loid)s and the kernel density simulation of vegetable fields. The equations of the two techniques used were:

(1) Ordinary Kriging

According to the ordinary kriging interpolation technique, if the attribute value of variable *Z*(*x*) for study area *a* at point x_*i*_
∈ A (*i* = 1, 2, …, *n*) was $Z(x_{i})$, then the kriging interpolation result of the attribute value $Z^{*}(x_{0})$ at the to-be-interpolated point $x_{0}$
∈ A was the weighted sum of the attribute values $Z(x_{i})$(*i* = 1, 2, …, *n*) of the most neighboring known sample points, namely:(1)$$Z^{*}(x_{0})=\sum_{i=1}^{n}\lambda_{i}Z(x_{i})$$
where $\lambda_{i}$ is the weight assigned to each $Z(x_{i})$ value, with their sum being 1, and *n* is the amount of the most neighboring sampled data points used for the estimation.

(2) Kernel Density

Kernel density estimation is a particularly useful method of density estimation. A set of data can be used to continuously replace discrete histograms to create a smooth curve. The universal equation of kernel density estimation is mathematically expressed as follows:(2)$${\overset{\wedge }{p}}_{n}(x)=\frac{1}{nh}\sum_{1}^{n}k\left(\frac{X_{i}-x}{h}\right)$$
where k(x) is the kernel function, which is usually a smooth symmetric function, such as a Gaussian function, and *h* is the smoothing bandwidth if it is greater than 0, which controls the amount of smoothing. Kernel density estimation smooths each data point $X_{i}$ into a small density bump and then adds all of the small bumps to obtain a final density estimate.

### 2.4. Statistical Analyses

Statistical analyses were performed using SPSS 19.0 (International Business Machines Corporation, Armonk, NY, USA) [28]. Pearson’s correlation analysis was used to measure the linear relationship between the two variables by detecting if two phenomena (statistics) are correlated. In this study, Pearson’s correlations were performed to verify if the distributions of the soil metal(loid)s were closely related to the locations of the vegetable fields.

### 2.5. Spatial Autocorrelation Analysis

Moran’s I is the most commonly used method for calculating spatial autocorrelation, both global and local spatial autocorrelations [1,29,30].

The global Moran’s I is derived from the covariance relationship between correlation coefficients. The value of the covariate also represents the correlation between the two sets of values. The global Moran’s I is expressed as follows:(3)$$I=\frac{n}{\sum_{i=1}^{n}\sum_{j=1}^{n}Wji}\times \frac{\sum_{i=1}^{n}\sum_{j=1}^{n}Wij\left(x_{i}-\bar{x}\right)\left(x_{j}-\bar{x}\right)}{\sum_{i=1}^{n}{\left(x_{i}-\bar{x}\right)}^{2}}$$
where *W**ij* is the spatial adjacency weight matrix of each spatial unit *i* and the spatial unit *j* (j={1,2,3,…,n}) in the study area (1 indicates that *i* is adjacent to *j*, while 0 indicates that *i* and *j* are not adjacent), $x_{i}$ is the value of each variable in a variable set, with their average being $\bar{x}$, and $y_{i}$ is the value of each variable in another variable set, with their average being $\bar{y}$.

The local Moran’s I is a local measure of spatial autocorrelation, which is used to identify the locations of spatial clusters and spatial outliers. It is calculated as follows:(4)$$I_{i}=\frac{n\left(x_{i}-\bar{x}\right)\sum_{j=1}^{n}Wij\left(x_{j}-\bar{x}\right)}{\sum_{i=1}^{n}{\left(x_{i}-\bar{x}\right)}^{2}}$$

The definition of each variable is similar to that provided for the above equation. The Moran’s I value calculated according Equation (4) is between −1 and 1. If the value is greater than 0, the correlation is positive. If the value is less than 0, the correlation is negative. A larger value indicates a greater spatial distribution correlation, i.e., the spatial distribution is clustered. A smaller value indicates a small spatial distribution correlation. When the value tends to be 0, the spatial distribution is random.

The global and local Moran’s I were analyzed in GeoDa [31], which is a user-friendly software with a wide set of spatial analysis methods, by which the global Moran’s I value and its significance can be obtained, as well as the local spatial autocorrelation classification results of local Moran’s I statistical analysis.

### 2.6. Technical Path

A geographic polygon database was created in ArcMap (Environmental Systems Research Institute, Inc, RedLands, CA, USA). The data for the metal(loid)s in soils, including the soil metal(loid) concentrations and latitude/longitude information, were imported into the database. The Kernel Density tool in ArcMap was used to obtain the kernel density estimate of vegetable field data in the study area. A 1 × 1 km grid was created for the entire study area. The join association method in ArcGIS was then used to correlate the metal(loid) concentrations with the kernel density of vegetable fields based on spatial locations, and the correlated grid data were finally exported. Then, a spatial weight matrix was constructed in GeoDa (GeoDa Center for Geospatial Analysis in the University of Chicago, Chicago, IL, USA), and the bivariate local Moran’s I was used to generate a cluster map. The generated type results were saved to the data file for subsequent operations and analyses in ArcGIS.

## 3. Results and Discussion

### 3.1. Geostatistical Analysis of the Spatial Patterns of Metal(loid)s

#### 3.1.1. Variations and Correlations of Soil Heavy Metals

The analysis of the raw sampling data (Table 2) indicated that the coefficient of variation of soil Hg was the highest (85.597%), followed by soil Cd (41.804%), and soil As (35.081%), which may be related to human activities. The variation intensity of soil pH was the lowest. The coefficient of variation can only qualitatively reflect the overall level of soil attributes and their changing trends but cannot reflect changes in their spatial characteristics. Neither can it quantitatively describe how soil attributes change with sampling sites or determine whether structural or random factors are the most influential on variation. Semivariogram features are needed to explain the above characteristics.

The nugget effect, one of the features of the semivariogram, is the ratio of the nugget value to the sill value, i.e., C0/(C0 + C), which indicates the proportion of the spatial variability due to randomness in the total variation of the system. It falls into three grades according to the spatial correlation degrees of the region-specific variables. If the nugget coefficient is <25%, the spatial correlation is strong. If the nugget coefficient is 25–75%, the spatial correlation is moderate. If the nugget effect is >75%, the spatial correlation is weak. It can be seen from Table 3 that the soil As showed a strong spatial correlation, while the other soil attributes showed a moderate spatial correlation.

#### 3.1.2. Spatial Distribution of Metal(loid)s—Kriging Model Interpolation

The concentrations of the five metal(loid)s examined here (Table 2) were compared with the soil pollution risk values of agricultural lands published in China’s Environmental Quality Standard for Soils [32]. The average concentrations of the five metal(loid)s did not exceed Grade II soil quality. However, the concentrations of the five metal(loid)s exceeded the soil background value in some areas, indicating that the farmland soils in the study area may have been contaminated by metal(loid)s to different degrees. Combining the results of the spatial distribution of vegetable fields with that of metal(loid)s, it can be depicted that the locations of the vegetable fields were closely related to soil pH and heavy metal spatial distributions.

For the purpose of understanding the spatial distribution regularities of the different metal(loid)s, the kriging interpolation method of the geostatistical module in ArcGIS was used to assess the metal(loid) concentrations in each type of soil in the study area. Based on this, a spatial distribution map of the heavy metal concentrations in the soil was created (Figure 4). The spatial distributions of Cr, Pb, Cd, As, and Hg, as well as the pH of sampled areas showed an increasing or decreasing trend, and all these attributes had obvious high-value areas. All metal(loid)s showed high values in the south-eastern area, possibly due to phosphate mining. The concentrations of Cr, Pb, and Hg were generally low in the study area. While Cr was clustered in several small sites of the north-eastern area, possibly due to local paint factories, the concentration of As was high in the north-western area possibly due to building material and paint factories. High Hg concentrations were found in the south-eastern area, mainly due to mining plants, and Cd was mainly distributed in the southern area due to cities and mines, and in the north-western area due to agriculture, especially vegetable fields. Lead and Cr were mainly distributed in the western area, possibly due to agricultural production rather than industry and mining. Since long-term vegetable planting caused soil acidification, high pH values (Figure 5) were found to be negatively correlated with the spatial distribution of vegetable fields (Figure 4).

#### 3.1.3. The Vegetable Field Kernel Density

The probability density of different vegetable field types in the study area was obtained by kernel density analysis method. The centroid of each polygon was extracted as the point data for the kernel density calculation. The default search radius (bandwidth) was calculated based on the spatial configuration and number of input points. The bandwidth was calculated by dividing the minimum value of the width or the height of the output range in the study area by 30. For example, the east-west width of the study area is 47,670 m, and the default bandwidth is 1589. This approach to calculate the default radius generally avoids the “ring around the points” phenomenon that often occurs with sparse datasets. This approach corrects for spatial outliers—input points that are very far away from the rest—so they will not make the search radius unreasonably large. The results show that both vegetable fields have spatial agglomeration characteristics, mostly around villages and towns (Figure 6).

### 3.2. Statistical Analyses

#### 3.2.1. Correlation Analysis

The effects of the vegetable fields on the geochemical characteristics of soil metal(loid)s and pH were analyzed, but the normality test did not pass for the Kolmogorov-Smirnova test and the Shapiro-Wilk test in Table 4. Thus, the original data were transformed by the bloom method and then were assessed by examining Pearson’s correlations between metal(loid) interpolation data and vegetable field kernel density in the 2276 1 × 1 km grids across the study area.

The correlation coefficients are listed in Table 5. The *p*-values lower than 0.01 evidenced the significant correlation between soil Cr, Pb, Cd, As, Hg concentrations, and pH with the density of OV and GV fields. Since the planting times of GV fields varied greatly, there was no obvious regularity of their effects on metal(loid) concentrations. Open field vegetables showed high significant correlations with soil attributes due to their long-term planting history, and these were positive for Cd and Hg, and negative for Cr, Pb, As, and pH. This was primarily because in OV fields, the massive use of organic fertilizers resulted in bioconcentrations of Cd and Hg, and long-term fertilization significantly reduced the pH, leading to the transformation and migration of Cr, Pb, and As.

#### 3.2.2. Heavy Metal Concentrations on the Different Types of Soil and Vegetable Fields

To analyze whether heavy metal accumulation in the soil was related to the soil type and vegetable field type, the ArcGIS statistical data tool (Zonal Statistics as Table) was used to superimpose the soil type, vegetable field type, and heavy metal grid data to obtain the number, mean, standard deviation, etc., of pixels in the type-specific statistically framed areas.

Table 6 and Figure 7 show the average concentrations of metal(loid)s in the different types of soils. The average concentrations of all metal(loid)s were low and did not reach the soil pollution background values of Anhui. However, the heavy metal concentrations in the different types of soils varied significantly, which was related to the soil background values. In the limestone and skeletal soils, the concentrations of the four metal(loid)s other than Cd were high. In paddy soils, the concentration of Cd was high, while that of the other four metal(loid)s were low, which may be related to soil fertilization characteristics.

Table 7 and Figure 8 show the average concentrations of metal(loid)s in the different vegetable fields. The average pH and Cd concentrations of OV fields were lower than those of the GV fields, whereas the average As, Cr, Hg, and Pb concentrations of the OV fields were higher than those of the GV fields. This was possibly because the long planting history of OV fields led to soil acidification (low pH), and because OV fields are susceptible to atmospheric deposition of metal(loid)s. Therefore, the As, Cr, Hg, and Pb concentrations of the OV fields were higher. The average concentration of Cd in the GV fields was high, possibly because the amount of applied organic fertilizer in such fields far exceeded that of the OV fields and resulted in Cd accumulation in the soils.

The pH of the GV fields increased with the planting area (Table 8), which was related to the latest compensation policy for greenhouse vegetable cultivation in Feidong, as described below. The compensation standards vary with planting areas and the larger the planting area, the higher the compensation. Compensation is given according to five levels of planting areas, i.e., 1 (0–1.3 ha), 2 (1.3–3.3 ha), 3 (3.3–5 ha), 4 (5–6.6 ha), and 5 (≥6.6 ha). Therefore, some large-scale GV fields have emerged in recent years, but their soils are not severely damaged, and their pH is high due to the short planting time. Level 1 GV fields were characterized by small planting areas and long planting times, and vegetables were mostly for the collective consumption of a family or village, so the soil pH was the lowest, posing a certain risk of migration. Level 2 and 3 GV fields used more fertilizer as they produced vegetables for sale, resulting in higher metal(loid) concentrations in the soils. However, the average metal(loid) concentrations varied with the use of different organic fertilizers. For Level 4 GV fields, the planting area was large, and the planting time was short; the use of organic fertilizers increased the concentrations of As and Cd, and reduced Pb accumulation in soils.

No regularity was found in OV fields, which was possibly related to the size of the investigated area and to differences in the fertilization habits. Level 1 OV fields were characterized by small planting areas and had the highest pH. Their Cd and Hg concentrations were also the highest, probably because the small vegetable fields, located on the roadsides for the convenience of vegetable growers, were susceptible to automobile exhaust and atmospheric conditions. Level 5 OV fields were large, and had the lowest pH, as well as the lowest Pb and Hg concentrations. Level 3 OV fields had the highest As concentrations and level 4 OV fields had the highest Pb and Cr concentrations (Table 9).

Open vegetable fields had lower soil pH and higher soil As, Cr, and Hg concentrations than GV fields in the same area. Only OV fields of levels 2, 3, and 4 had lower Cd and Pb than GV fields in the same area. Heavy metal pollution in the OV fields was more severe than that in the GV fields. Moreover, OV fields were subject to the external environment, resulting in the loss of metal(loid)s (Figure 9).

### 3.3. Spatial Autocorrelation Analysis of OV Fields and Spatial Pattern Analysis of Metal(loid)s

#### 3.3.1. Spatial Autocorrelation Comparison between OV and GV

The global Moran’s I describes the overall distribution of a phenomenon and determines whether this phenomenon has a spatial clustering characteristic. Generally, at the 5% significance level, if Z(I) is greater than 1.96, it indicates that the distribution of the phenomenon has significant spatial autocorrelation. If Z(I) is less than −1.96, it indicates that the distribution of the phenomenon within the study area has negative spatial autocorrelation. The global Moran’s I values found here revealed that the five metal(loid)s and pH had significant spatial autocorrelations (all *p* < 0.01). The global Moran’s I values were ranked in the following sequence: Pb > As > pH > Cr > Hg > Cd (Table 10).

The bivariate local Moran’s I is a measure of the spatial autocorrelation between two variables. Our results showed that the five metal(loid)s and pH had significant spatial autocorrelations (all *p* < 0.01), as presented in Table 11 and Table 12. When the absolute bivariate local Moran’s I of each soil attribute were compared, OV fields showed greater correlations than GV fields, possibly because the latter did not witness an apparent regularity in time and space. Open vegetable fields had a positive spatial correlation with soil Hg and Cd concentrations, and a negative spatial correlation with soil Pb, As, and Cr concentrations, possibly because the reduction in pH accelerated the loss of the latter three metal(loid)s after activation.

#### 3.3.2. The Sensitivity Analysis of the Modifiable Areal Unit Problem (MAUP)

The modifiable area unit problem (MAUP), a term introduced by Openshaw and Taylor’s classic paper [33], has long been recognized as a potentially troublesome feature of aggregated data. The deviation between the statistical results and the spatial analysis results caused by the change of the aggregated spatial units is often described by scale effect and zoning effect. In order to clarify the MAUP effect in this paper, three scales of geographical units were selected to analyze the modifiable areal unit problem sensitivities. The results of bivariate local Moran’s I value coefficients were calculated, as shown in Table 13, which was sensitive and depended on the specific geographic unit used in the study. The spatial autocorrelation coefficients of different geographic units were different, but there was a scale (1000 × 1000 m grid) that could explain the spatial sensitivity reasonably. The other two scales, the positive correlation between pH and OV was obviously inconsistent with the survey results, and the spatial relationship between other metal(loid)s and OV was difficult to explain. Therefore, the follow-up analysis only analyzed the medium-scale (1 × 1 km).

Although the global Moran’s I described the overall distribution of metal(loid)s and their spatial clustering, it did not specify in which regions the metal(loid)s were clustered. On the other hand, the bivariate local Moran’s I distribution map produced (Figure 10) evidenced four types of local spatial correlations of variables in each region and surrounding regions:(1)Areas with high Moran’s I for both soil metal(loid)s and OV fields (high-high areas): OV fields were highly clustered and witnessed severe accumulation of metal(loid)s, showing a certain clustering scale effect.(2)Areas with high Moran’s I for soil metal(loid)s and low Moran’s I for OV fields (high-low area): OV fields were lowly clustered and witnessed severe accumulation of metal(loid)s. The concentrations of metal(loid)s in soils were extremely high, possibly due to non-agricultural production or other environmental factors such as soil parent material.(3)Areas with low Moran’s I for soil metal(loid)s and high Moran’s I for OV fields (low-high area): OV fields were highly clustered and witnessed no accumulation of metal(loid)s.(4)Areas with low Moran’s I for both soil metal(loid)s and OV fields (low-low area): OV fields were lowly clustered and witnessed a low accumulation of metal(loid)s.(5)As shown in Figure 10, the results of spatial autocorrelation distribution of variables were quite different due to different geographical units. The spatial autocorrelation of smaller geographical units was much stronger than that of larger geographical units. Table 13 also shows that the coefficient of bivariate local Moran’s I values decreased with the increase of larger spatial units. The sensitivity analysis of the modifiable area unit problem illustrates the uncertainty caused by the scale effect, and different scales may lead to some information loss or bias.

As the bivariate local Moran’s I values of the GV fields were low, this study only analyzed the bivariate local Moran’s I distribution map of the OV fields (Figure 11). Their spatial distribution characteristics showed that OV fields had different effects on the spatial distributions of the six soil attributes. Concentrations of Pb and Cr and pH were highest in the soils of high–low areas, among which the pH was the most significant. Open vegetable fields were lowly clustered, and their soil pH was high, probably because vegetable planting in OV fields removed excessive base elements, thereby leading to soil acidification. Concentrations of Pb and Cr showed a negative spatial correlation, probably because the humus resulting from organic fertilizer decomposition caused soil acidification to increase the activity of Pb and Cr in soils, thereby resulting in their loss via mobilization. Concentrations of Hg and Cd were highest in the soils of low-low areas and showed no obvious negative spatial correlations. Farmers in Feidong County used to apply organic fertilizers such as chicken manure and cow dung to grow vegetables, generally 18,750 kg/hectares for common vegetables and tomatoes and 15,000 kg/hectares for lettuce, cabbage, Hangzhou pepper, and pepper. Such organic matter had a certain reduction ability that enabled Hg and Cd in soil solutions to form sulfides and precipitate, so their migration was less important than that of Pb and Cr. The five metal(loid)s presented low values in soils of high–high areas, i.e., the cumulative effects of OV fields and metal(loid)s were not obvious, possibly because the bioconcentration of metal(loid)s in the soils was affected by many factors, such as the application of fertilizers and pesticides, agricultural planting structure, sewage irrigation, soil properties, different climatic conditions, and atmospheric deposition. Thus, OV fields alone cannot explain all the spatial variability of metal(loid) bioconcentration.

## 4. Conclusions

By combining GIS and Moran’s I spatial analysis to generate spatial autocorrelation distribution maps, the spatial clustering (positive autocorrelation) and spatial anomaly (negative autocorrelation) of vegetable fields and soil heavy metal concentrations were identified. It can be seen from GIS spatial analysis that the spatial distributions of Cr, Pb, Cd, As, and Hg content and pH in the study area showed a certain increasing or decreasing trend, and all showed spatial clustering to some extent. It can be concluded that even though the background values of metal(loid)s in the soils of Feidong are low, anthropogenic activities have contributed to an alarming increase of metal(loid) concentrations. On this basis, spatial correlation analysis was performed to assess the effects of vegetable fields on metal(loid) accumulation in soils, which is important for protecting soil quality and improving crop quality. Long-term vegetable production in the study area involves the massive use of organic fertilizers, which affects the spatial distribution of metal(loid)s in soils, resulting in the bioconcentration of Cd and Hg. Long-term fertilization significantly reduced pH, thereby causing transformation and migration of Cr, Pb, and As. The soil pH of the GV fields increased with the planting area, which was related to the vegetable compensation policy in the study area. The planting history is short, but the accumulation rate of Cd is faster, which has a certain migration risk. Open vegetable fields, with a long planting history, had lower than average soil pH and Cd concentration than GV fields, and higher average soil As, Cr, Hg, and Pb concentrations than GV fields. Metal(loid)s in soil samples caused potential hazards through non-point source pollution transfer. Therefore, special attention should be paid to the type and quantity of fertilizer in order to reduce the transformation and loss of metal(loid)s in soils.

In the future, the three-dimensional spatial variability of metal(loid)s in soil will be focused on. Through the study of metal(loid) migration and transformation in vertical and horizontal directions, we can clarify the environmental impact of metal(loid)s in OV and GV to the groundwater and surrounding ditches respectively.

## Figures and Tables

**Figure 1 ijerph-16-04095-f001:**
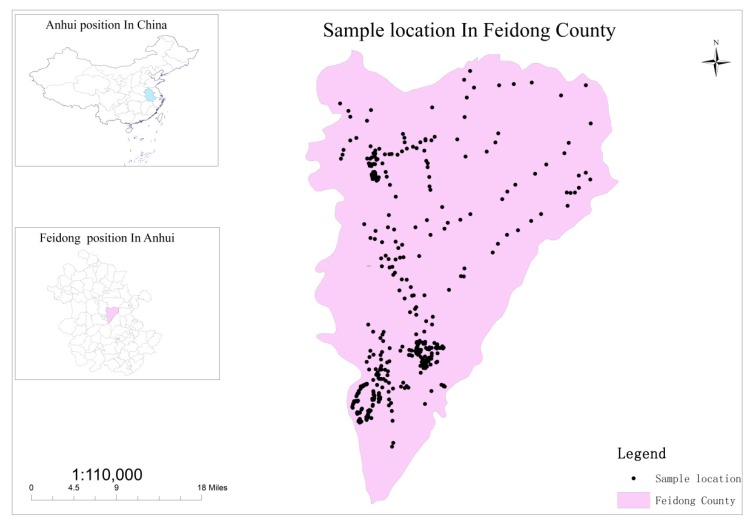
Relative location of Feidong County in Anhui Province, China, and distribution of the sampling sites in Feidong County.

**Figure 2 ijerph-16-04095-f002:**
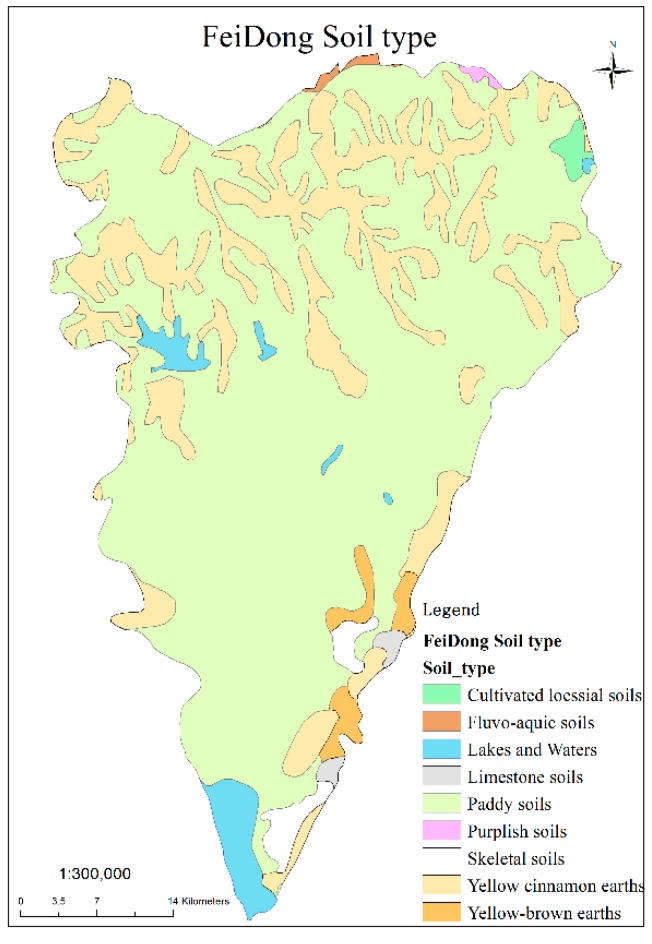
Distribution of the soil types in Feidong County (data from Anhui Provincial Agriculture Committee).

**Figure 3 ijerph-16-04095-f003:**
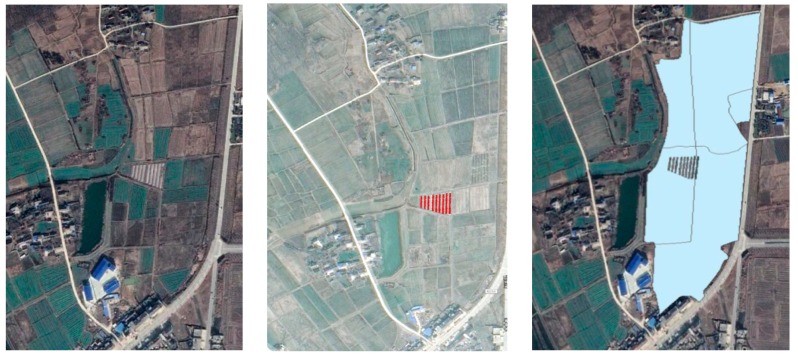
Spatial extraction of different types of vegetable fields.

**Figure 4 ijerph-16-04095-f004:**
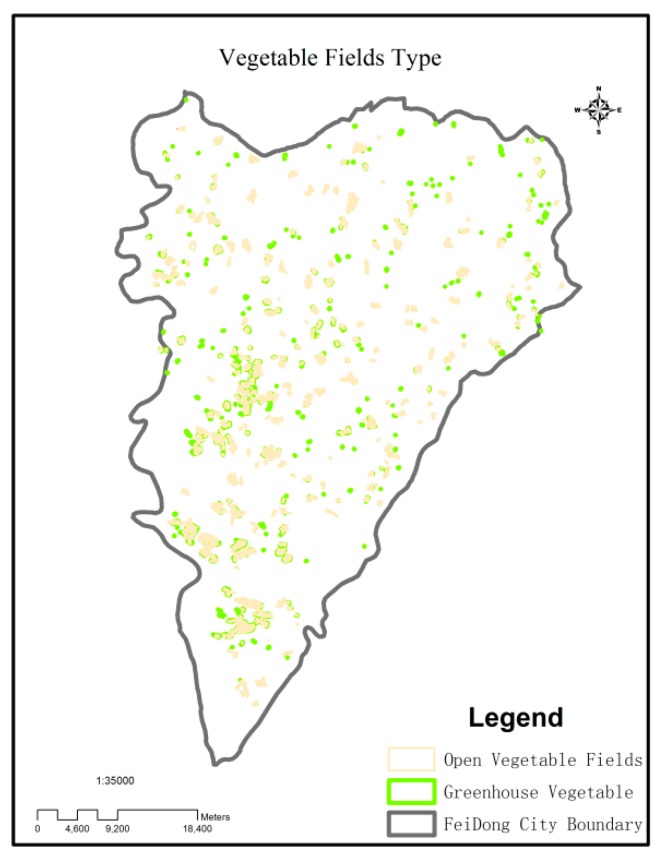
Spatial distributions of different types of vegetable fields.

**Figure 5 ijerph-16-04095-f005:**
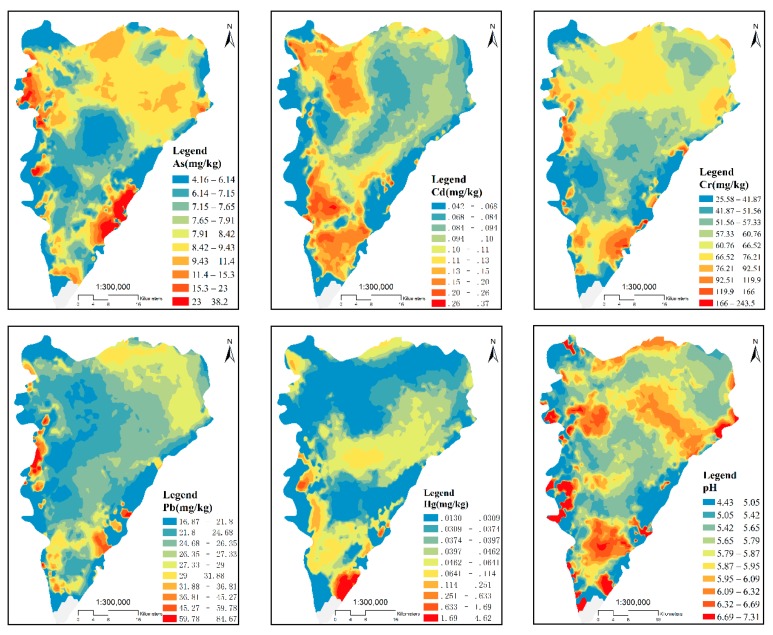
Spatial distributions of metal(loid) concentrations.

**Figure 6 ijerph-16-04095-f006:**
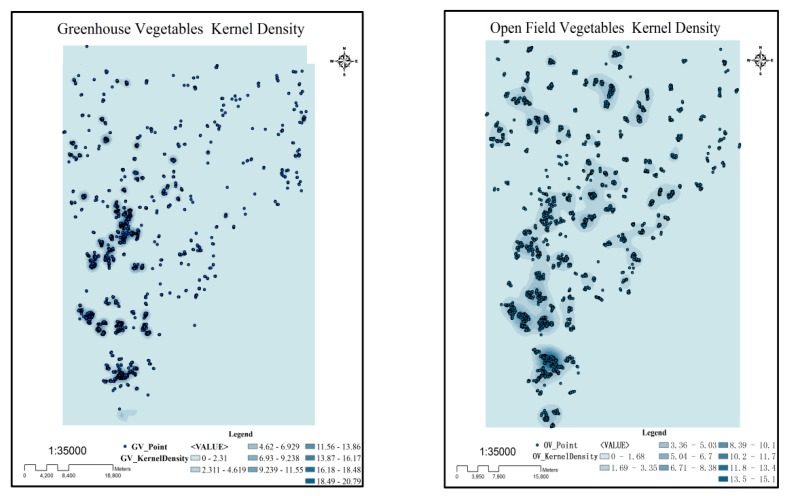
The kernel density spatial distributions of greenhouse vegetables (GV) and open field vegetables (OV).

**Figure 7 ijerph-16-04095-f007:**
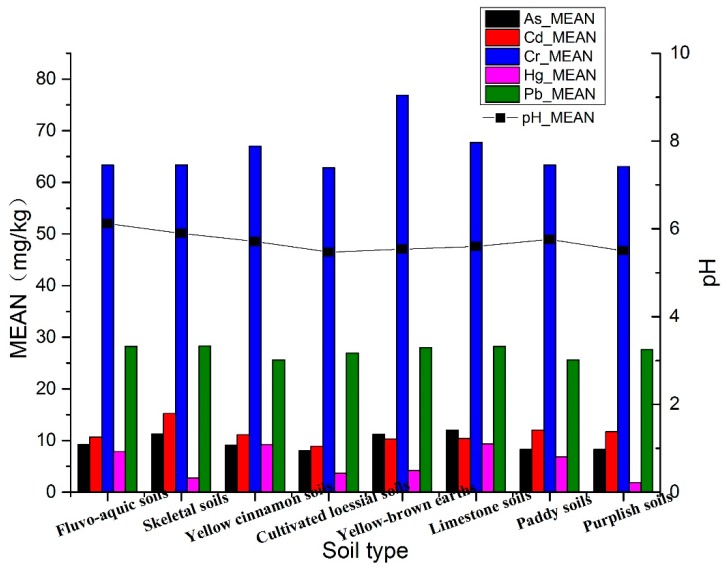
Average concentrations of metal(loid)s in the different types of soils (shown at a uniform scale; Cd and Hg concentrations are magnified 100 times in the figure).

**Figure 8 ijerph-16-04095-f008:**
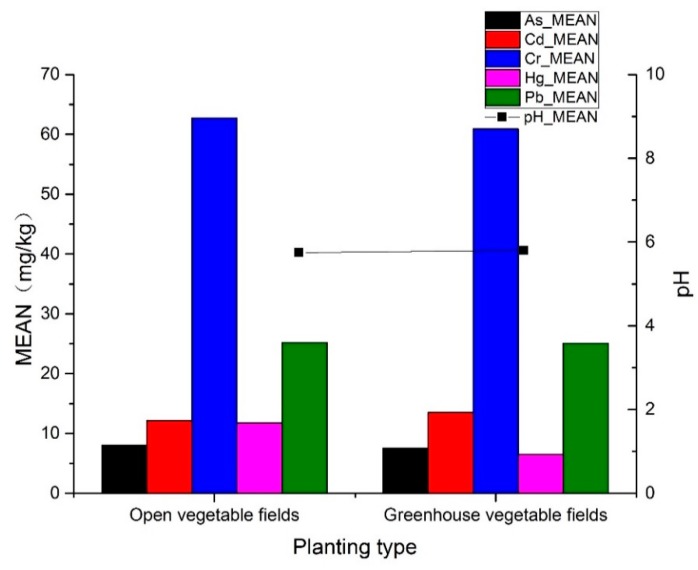
Average concentrations of metal(loid)s and pH in the different vegetable fields (shown at a uniform scale; Cd and Hg concentrations are magnified 100 times in the figure).

**Figure 9 ijerph-16-04095-f009:**
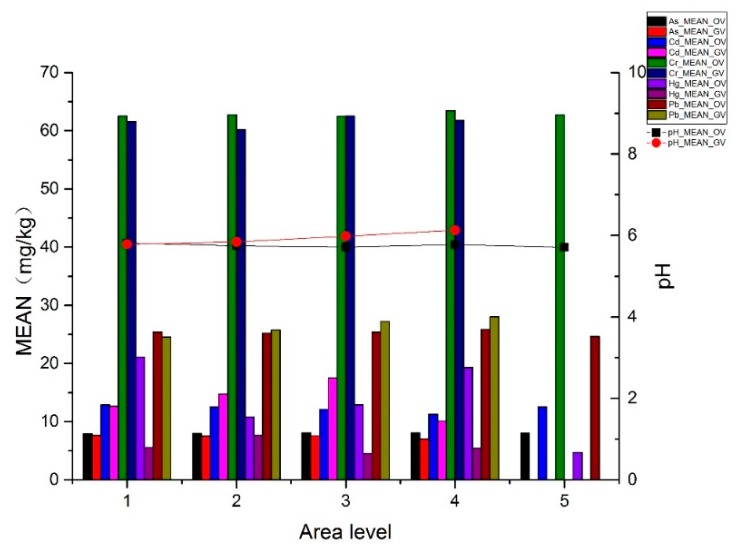
Average concentrations of metal(loid)s at the open and greenhouse vegetable fields (OV and GV, respectively) according to area levels (shown at a uniform scale; Cd and Hg concentrations are magnified 100 times in the figure).

**Figure 10 ijerph-16-04095-f010:**
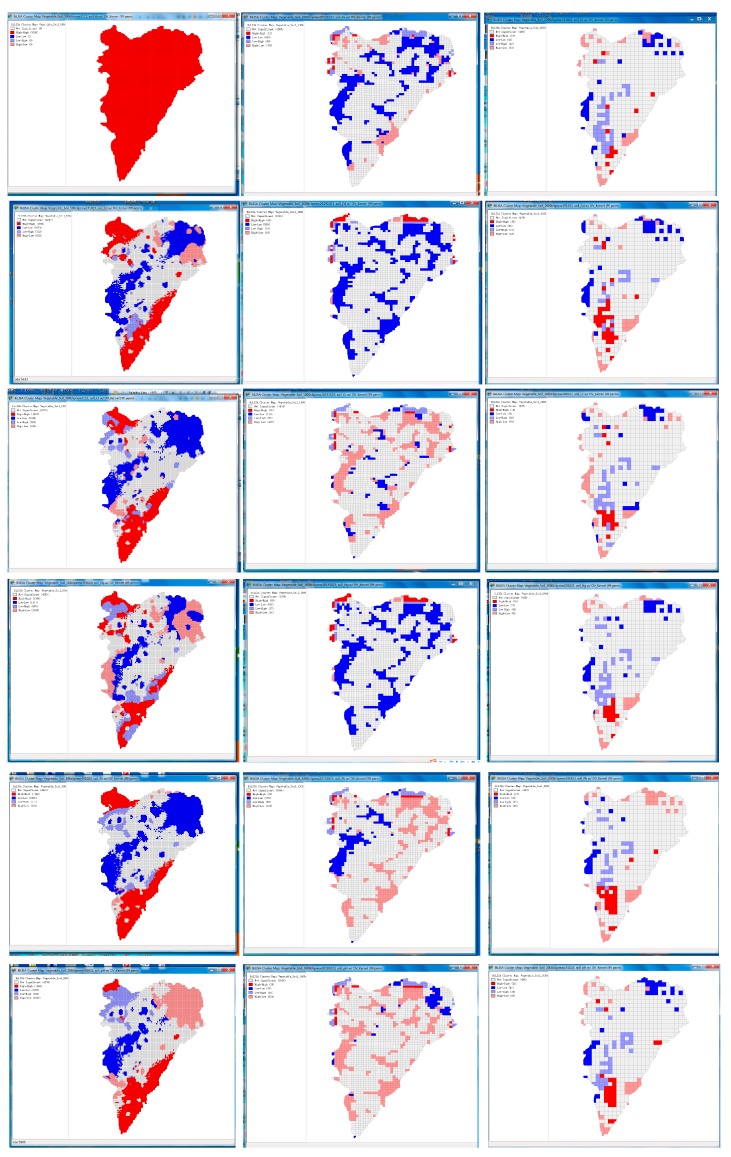
Bivariate local Moran’s I distribution maps for metal(loid)s and pH in different GRID (Left 500 m, Middle 1000 m, Right 2000 m, and Rows 1 to 6 were As, Cd, Cr, Hg, Pb, and pH).

**Figure 11 ijerph-16-04095-f011:**
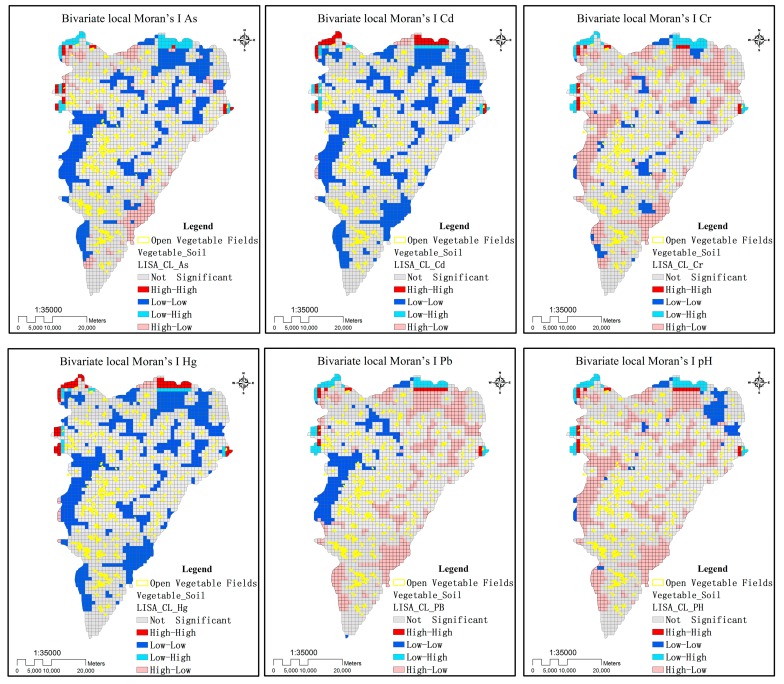
Bivariate local Moran’s I distribution maps for metal(loid)s and pH by 1 × 1 km Unit.

**Table 1 ijerph-16-04095-t001:** Importance of the soil types in Feidong County (data from Anhui Provincial Agriculture Committee).

Soil Type	Area (km^2^)	Percentage
Paddy soils	1653	72.72
Yellow cinnamon soils	462	20.33
Lakes and waters	64	2.82
Yellow–brown earths	34	1.50
Skeletal soils	31	1.36
Cultivated loessial soils	11	0.48
Limestone soils	10	0.44
Fluvo-aquic soils	5	0.22
Purplish soils	3	0.13

**Table 2 ijerph-16-04095-t002:** Analysis of raw sampling data.

Soil Attribute	Minimum Value (mg/kg)	Maximum Value (mg/kg)	Average Value (mg/kg)	Median (mg/kg)	Standard Deviation	Coefficient of Variation (%)	Skewness	Kurtosis
As	4.161	38.238	8.262	7.852	2.898	35.081	5.418	50.906
Cd	0.042	0.374	0.128	0.114	0.053	41.804	1.158	4.479
Cr	25.377	243.460	65.027	62.460	17.617	27.092	3.586	31.975
Hg	0.013	4.619	0.059	0.051	0.050	85.597	7.858	340.520
Pb	16.870	84.668	27.190	26.697	5.468	20.108	3.897	37.165
pH	4.430	7.231	5.887	5.870	0.481	8.174	0.105	3.277

Samples (*n* = 375).

**Table 3 ijerph-16-04095-t003:** Semivariogram parameters for soil attributes.

Soil Attribute	Range (m)		Long-Axis Angle (°)	Nugget (C_0_)	Structural Variance (C)	Sill (C_0_ + C)	Nugget/Sill (%)	Root-Mean-Square Deviation	Average Standard Deviation
	Long Axis	Short Axis							
As	1849.340	815.360	76.640	0.007	0.035	0.042	16.667	1.044	1.852
Cd	2498.280	1778.250	6.150	0.035	0.098	0.133	26.316	0.964	0.034
Cr	2400.770	1709.370	136.230	0.026	0.025	0.050	50.996	0.920	11.672
Hg	2180.640	1237.300	112.140	0.050	0.133	0.183	27.322	1.366	0.057
Pb	528.920	393.810	13.530	0.006	0.013	0.019	31.579	1.286	4.491
pH	8841.320	425.870	55.540	0.076	0.097	0.173	43.931	1.087	0.390

Samples (*n* = 375).

**Table 4 ijerph-16-04095-t004:** Normality test.

Title	Kolmogorov-Smirnova			Shapiro-Wilk		
	Statistics	df	Sig.	Statistics	df	Sig.
OV_KernelDensity	0.513	2276	0	0.075	2276	0
GV_KernelDensity	0.333	2276	0	0.471	2276	0
As	0.329	2276	0	0.333	2276	0
Cd	0.532	2276	0	0.2	2276	0
Cr	0.232	2276	0	0.679	2276	0
Hg	0.511	2276	0	0.233	2276	0
Pb	0.251	2276	0	0.584	2276	0
pH	0.378	2276	0	0.431	2276	0

Samples (*n* = 2276).

**Table 5 ijerph-16-04095-t005:** Correlation analysis between the kernel density and soil heavy metal concentrations on open and greenhouse vegetable fields (OV and GV, respectively).

	OV	GV	As	Cd	Cr	Hg	Pb	pH
OV	1	−0.042 *	−0.113 **	0.474 **	−0.384 **	0.468 **	−0.334 **	−0.385 **
GV	−0.042 *	1	−0.065 **	−0.089 **	0.014	−0.098 **	0.055 **	0.105 **
As	−0.113 **	−0.065 **	1	−0.234 **	−0.068 **	−0.082 **	−0.409 **	−0.316 **
Cd	0.474 **	−0.089 **	−0.234 **	1	−0.803 **	0.983 **	−0.696 **	−0.806 **
Cr	−0.384 **	0.014	−0.068 **	−0.803 **	1	−0.842 **	0.815 **	0.850 **
Hg	0.468 **	−0.098 **	−0.082 **	0.983 **	−0.842 **	1	−0.786 **	−0.883 **
Pb	−0.334 **	0.055 **	−0.409 **	−0.696 **	0.815 **	−0.786 **	1	0.925 **
pH	−0.385 **	0.105 **	−0.316 **	−0.806 **	0.850 **	−0.883 **	0.925 **	1

Samples (*n* = 2276); * indicates a significant relationship at *p* < 0.05. ** indicates a significant relationship at *p* < 0.01.

**Table 6 ijerph-16-04095-t006:** Average concentrations of metal(loid)s and pH in the different types of soils.

Type	pH	As (mg/kg)	Cd (mg/kg)	Cr (mg/kg)	Hg (mg/kg)	Pb (mg/kg)
Cultivated loessial soils	5.4704	8.0111	0.0890	62.7960	0.0362	26.9889
Fluvo–aquic soils	6.1199	9.2413	0.1071	63.3214	0.0784	28.2519
Limestone soils	5.5975	12.0172	0.1043	67.7190	0.0931	28.1897
Paddy soils	5.7602	8.2770	0.1203	63.3104	0.0683	25.6234
Purplish soils	5.5017	8.2490	0.1173	63.0286	0.0187	27.6239
Skeletal soils	5.8965	11.2538	0.1525	63.3785	2.0277	28.2820
Yellow cinnamon soils	5.7111	9.1123	0.1112	67.0106	0.0917	25.6263
Yellow-brown earths	5.5385	11.1875	0.1032	76.8729	0.0423	28.0242

**Table 7 ijerph-16-04095-t007:** Average concentrations of metal(loid)s and pH in the open and greenhouse vegetable fields (OV and GV, respectively).

Type	pH	As (mg/kg)	Cd (mg/kg)	Cr (mg/kg)	Hg (mg/kg)	Pb (mg/kg)
OV	5.7467	8.0250	0.1215	62.7534	0.1175	25.1793
GV	5.8005	7.5851	0.1352	60.9272	0.0650	25.0702

**Table 8 ijerph-16-04095-t008:** Average metal(loid) concentrations of greenhouse vegetable fields with different planting areas.

Area Class	pH	As (mg/kg)	Cd (mg/kg)	Cr (mg/kg)	Hg (mg/kg)	Pb (mg/kg)
1	5.788	7.640	0.127	61.591	0.056	24.533
2	5.840	7.527	0.147	60.202	0.077	25.760
3	5.985	7.612	0.175	62.507	0.046	27.245
4	6.128	6.996	0.101	61.787	0.055	27.651

Area classes are as mentioned in the text.

**Table 9 ijerph-16-04095-t009:** Average metal(loid) concentrations of open vegetable fields with different planting areas.

Area Class	pH	As (mg/kg)	Cd (mg/kg)	Cr (mg/kg)	Hg (mg/kg)	Pb (mg/kg)
1	5.813	7.922	0.129	62.542	0.211	25.427
2	5.747	7.981	0.125	62.752	0.108	25.247
3	5.720	8.094	0.121	62.478	0.129	25.405
4	5.774	8.086	0.113	63.463	0.193	25.850
5	5.713	8.070	0.125	62.777	0.047	24.652

Area classes are as mentioned in the text.

**Table 10 ijerph-16-04095-t010:** Global Moran’s I values for the metal(loid)s and pH.

Soil Attribute	Global Moran’s I	Z Value	*p*-Value
As	0.765	70.348	0.001
Cd	0.563	53.043	0.001
Cr	0.737	69.170	0.001
Hg	0.623	58.404	0.001
Pb	0.794	73.554	0.001
pH	0.748	69.764	0.001

**Table 11 ijerph-16-04095-t011:** Bivariate local Moran’s I values between the different metal(loid)s and open vegetable fields.

Soil Attribute	Bivariate Local Moran’s I	Z Value	*p* Value
As	−0.261	−31.294	0.001
Cd	0.258	31.180	0.001
Cr	−0.244	−30.163	0.001
Hg	0.277	33.734	0.001
Pb	−0.236	−29.737	0.001
pH	−0.261	−31.572	0.001

**Table 12 ijerph-16-04095-t012:** Bivariate local Moran’s I values between the different metal(loid)s and greenhouse vegetable fields.

Soil Attribute	Bivariate Local Moran’s I	Z Value	*p* Value
As	−0.075	−9.008	0.001
Cd	−0.065	−8.543	0.001
Cr	0.000	−0.054	0.001
Hg	−0.075	−9.780	0.001
Pb	0.038	5.168	0.001
pH	0.086	10.998	0.001

**Table 13 ijerph-16-04095-t013:** Bivariate local Moran’s I values of the different metal(loid)s and greenhouse vegetable fields from three scales.

Soil Attribute	Bivariate Local Moran’s I			Z Value			*p* Value
	500 × 500 m Grid	1000 × 1000 m Grid	2000 × 2000 m Grid	500 × 500 m Grid	1000 × 1000 m Grid	2000 × 2000 m Grid	
As	0	−0.261	−0.099	0	−31.294	−6.212	0.001
Cd	0.678	0.258	0.146	150.848	31.18	9.258	0.001
Cr	0.471	−0.244	−0.083	115.391	−30.163	−5.299	0.001
Hg	0.207	0.277	−0.006	53.765	33.734	−0.396	0.001
Pb	0.645	−0.236	−0.034	137.329	−29.737	−2.193	0.001
pH	0.162	−0.261	0.026	42.987	−31.572	1.6324	0.001

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
