# Peer review of "Effects of Vegetable Fields on the Spatial Distribution Patterns of Metal(loid)s in Soils Based on GIS and Moran’s I"

_ijerph, 2019, doi:10.3390/ijerph16214095_

Round 1

Reviewer 1 Report

The paper detailed effects of vegetable fields on heavy metal accumulation in soils. The GIS and RS methods are used in the study; however, the data processing and key parameters are not clear enough to the results. As a case study, I have the following comments:

The classification of the vegetable field types is not clear. The data source selection, classification methods, and classification accuracy analysis should be strengthened. How is the kernel density analysis method applied in the paper? In L234-236, the spatial distribution of heavy metals is generated by spatial interpolation; however, what is the meaning of “Geostatistical spatial interpolation was used to obtain the kernel density estimate of vegetable field data in the study area”(L196)? The key process, such as the calculation the vegetable field kernel density, and the spatial sampling design need be more detailed. The content of the sensitivity analysis of the paper, for example, the bandwidth selection of kernel density analysis, and the MAUP problem should be discussed. Some descriptions are not accurate, for example, “This study used spatial analysis methods combining Geographic Information Systems (GIS) and Moran’s I to analyze the effects of vegetable fields on heavy metal accumulation in soils.” The paper should further strengthen the explanation and discussion of the causes and mechanisms of the spatial distribution of the heavy metals.

Author Response

Dear reviewer:

Thank you very much for your valuable comments. We have corrected them one by one. Please see the attachment for details. Thanks again!

Best regards,

Youhua Ma

Reviewer 2 Report

The topic falls in the scope of International Journal of Environmental Research and Public Health and the authors have worked on very important question. Experiments seem well designed and methods used appropriated (except Pearson’s correlation analysis). Results are well described. However, there are still some issues that have to be addressed by the authors before considering the manuscript for publication. My comments are detailed below.

In the “Title”, “Abstract”, and throughout the manuscript, authors should review the use of the terms “heavy metal” and “metal”. For example, arsenic is not a heavy metal, or even a metal; arsenic is a metalloid. For example, authors can use the term “metal(loid)s” to represent both metals and metalloids. When authors want to refer specifically to heavy metals, and also want to include elements such as arsenic, then they should write “heavy metals and metalloids”.

Some minor typos, grammar and syntax errors should be carefully revised and corrected accordingly. For example: page 5, line 132, “heavy mental” should be “heavy metal”

Abstract

I suggest the authors to include some quantitative data which could be more interesting and informative for the readers.

Keywords

Authors should rephrase keywords. Do not use words or terms in the title as keywords: the function of keywords is to supplement the information given in the title. Words in the title are automatically included in indexes, and keywords serve as additional pointers.

Materials and Methods

In this section you should specify the characteristics of all equipments (report model, brand name, city and country of manufacturer).

Lines 162-167: I cannot understand the statistical analysis. The authors using parametric statistics (Pearson’s correlation analysis). Have the authors check for normality? Authors should explain which test they used for evaluation of the normality of the analysed features. It is known, for the scientist working on evaluation of pollutants in soil or sediment matrixes, that these substances never own normal distributions but highly skewed to the left and showing long right tails. Taking this into account I wonder they decided to use directly parametric statistics without (at least this is not noted in the manuscript) any previous evaluation of normality (e.g. Shapiro-Wilk test). For data not showing normal distributions there are a lot of equivalent statistical test that allow to do the same analysis but in a proper way.

Results and discussion

Tables and Figures: Authors should indicate the number of samples (n =).

Conclusions

The “Conclusions” section is quite long and it contains also discussion parts. I would suggest moving these discussions into the Discussion section and shortening of the Conclusions. Authors should avoid only repeating the results of the study, or present an abstract. The “Conclusions” section has a different meaning from the “Abstract” section. Authors should present here the main findings, including that which can contribute to new knowledge. Authors can also suggest possible implications and applications of this knowledge and “pathways” for future work.

Author Response

Dear Reviewer,

Thanks a lot for your valuable comments. We have corrected them one by one. Please see the attachment for details. Thanks again!

best regards,

Youhua Ma

Round 2

Reviewer 1 Report

1.The terms such as nuclear density should be revised according to the text book.

2.The sensitivity analysis of the paper with MAUP must be added to support the conclusion.

3.The paper should further strengthen the explanation and discussion of the causes and mechanisms of the spatial distribution of the heavy metals. It is missing in L450-460.

Author Response

(The authors gave the same response as above.)
